# Carboxyhemoglobin (COHb): Unavoidable Bystander or Protective Player?

**DOI:** 10.3390/antiox12061198

**Published:** 2023-05-31

**Authors:** André Carrola, Carlos C. Romão, Helena L. A. Vieira

**Affiliations:** 1UCIBIO, Applied Molecular Biosciences Unit, Department of Chemistry, NOVA School of Science and Technology, Universidade Nova de Lisboa, 2829-516 Caparica, Portugal; a.carrola@campus.fct.unl.pt; 2Instituto de Tecnologia Química e Biológica António Xavier, Universidade Nova de Lisboa, 2780-157 Oeiras, Portugal; ccr@itqb.unl.pt; 3Associate Laboratory i4HB—Institute for Health and Bioeconomy, NOVA School of Science and Technology, Universidade Nova de Lisboa, 2829-516 Caparica, Portugal

**Keywords:** hemoglobin, carbon monoxide, carboxyhemoglobin, oxidative stress, cytoprotection

## Abstract

Carbon monoxide (CO) is a cytoprotective endogenous gas that is ubiquitously produced by the stress response enzyme heme-oxygenase. Being a gas, CO rapidly diffuses through tissues and binds to hemoglobin (Hb) increasing carboxyhemoglobin (COHb) levels. COHb can be formed in erythrocytes or in plasma from cell-free Hb. Herein, it is discussed as to whether endogenous COHb is an innocuous and inevitable metabolic waste product or not, and it is hypothesized that COHb has a biological role. In the present review, literature data are presented to support this hypothesis based on two main premises: (i) there is no direct correlation between COHb levels and CO toxicity, and (ii) COHb seems to have a direct cytoprotective and antioxidant role in erythrocytes and in hemorrhagic models in vivo. Moreover, CO is also an antioxidant by generating COHb, which protects against the pro-oxidant damaging effects of cell-free Hb. Up to now, COHb has been considered as a sink for both exogenous and endogenous CO generated during CO intoxication or heme metabolism, respectively. Hallmarking COHb as an important molecule with a biological (and eventually beneficial) role is a turning point in CO biology research, namely in CO intoxication and CO cytoprotection.

## 1. Hemoglobin Function

Hemoglobin (Hb) is a widely conserved protein presented in all aerobic life forms where it is responsible for the transport and storage of oxygen (O_2_). In red blood cells (RBC), Hb corresponds to 98% of the total amount of cellular protein [1]. Hb is a globular protein with a tetrameric structure that is formed by two α and two β polypeptide chains. Each chain contains one heme group, a protoporphyrin IX molecule bound to reduced ferrous ion (Fe^2+^) which can bind one molecule of O_2_. Thus, Hb can bind up to four molecules of O_2_.

The primary biological role of Hb is the transport of O_2_ from the lungs to all tissues in the organism. Nevertheless, Hb also presents other functions related to its ability to bind other gaseous molecules, namely carbon dioxide (CO_2_), and the endogenous gasotransmitters: nitric oxide (NO), hydrogen sulfide (H_2_S), and carbon monoxide (CO) [2,3]. For the sake of completion, a brief overview of the main roles of the first three gasotransmitters is now presented, whereas the role of CO is more extensively presented in the next section.

CO_2_ is a metabolic product of the tricarboxylic acid cycle (TCA cycle) that affects pH, buffers the blood and inhibits innate immune and inflammatory responses [4]. Hb facilitates CO_2_ elimination from the organism, thus maintaining cellular homeostasis. Likewise, Hb affinity to O_2_ decreases in response to milieu acidification or to CO_2_-Hb binding which produces an allostery change. The decrease in Hb’s affinity to O_2_ is particularly important for facilitating O_2_ delivery into tissues [5].

NO is a gasotransmitter and an important signaling molecule enzymatically produced by the activity of nitric oxide synthase (NOS) or non-enzymatic reduction from nitrite. NO regulates vasomodulation by acting on vascular tone and on the expression of endothelial adhesion proteins, plays a role in immunity and inflammation, and is also a CNS signaling molecule [6,7]. Thus, the binding of Hb to NO alters its levels, changing cell signaling. In circulation, RBCs are the main scavengers of NO due to their high amounts of Hb. In fact, NO is much more reactive than CO, reacting with Hb in a much faster and efficient manner. OxyHb-Fe^2+^ converts NO into nitrate forming methemoglobin (MetHb or Hb-Fe^3+^) (Figure 1), while NO reacts with deoxyHb-Fe^2+^ forming DeoxyHb-Fe^+^ NO iron nitrosyl Hb [8,9].

Hydrogen sulfide (H_2_S) is a gasotransmitter and signaling molecule constantly produced by enzymes such as 3-MST (3-mercaptopyruvate sulfurtransferase), CBS (cystathionine β-synthase) and CSE (cystathionine γ-lyase). High concentrations of H_2_S are cytotoxic, but at low concentrations it possesses a variety of physiological functions, such as the regulation of blood vessel constriction and dilation and neuronal activity [10]. H_2_S can exclusively bind to methemoglobin (Hb-Fe^3+^) to form a methemoglobin–sulfide compound. Nevertheless, this compound can later undergo reductive sulfhydration resulting in the reduction of methemoglobin back to hemoglobin (Hb-Fe^2+^), along with the formation of polysulfides and thiosulfates. Although H_2_S mechanisms are not fully understood, the end goal of H_2_S-MetHb binding seems to be H_2_S transport and the capacity to reduce methemoglobin back to ferrous hemoglobin (Hb-Fe^2+^) [3].

CO is a key endogenous signaling gasotransmitter involved in inflammation, cell death, and metabolism control, maintaining cellular homeostasis, as described in the next section. Nevertheless, not much is known about the biological role of carboxyhemoglobin (COHb) under physiological conditions and its potential protective functions under pathological conditions.

## 2. Carbon Monoxide (CO) and Its Biological Role

As early as 1920, carbon monoxide (CO) was found in exhaled human air, a fact attributed to pollution and smoking and a bit later (1933) to metabolism of microbiota in the gut [11]. CO was identified as a product of heme catabolism only in 1951 [12], taking about two decades more for the identification of heme-oxygenase [13]. Heme-oxygenase (HO) can be inducible (HO-1) or constitutive (HO-2), and its expression and/or activity increases in response to stress along with CO production.

In the last two decades, CO has been extensively studied as a homeostatic and protective molecule. CO’s anti-inflammatory property was the first biological role to be scrutinized. In macrophages, CO controls inflammatory response [14,15] and the ability to kill bacteria [16]. Likewise CO also modulates and limits neuroinflammation in microglia [17,18] and in vivo models of multiple sclerosis [19,20]. CO plays an anti-apoptotic role in different cell types: endothelial cells [21,22]; lung cells [23]; cardiomyocytes [24]; neurons [25,26,27]; and astrocytes [28,29,30]. Another biological process regulated by CO is cell metabolism, namely the balance between glycolysis and oxidative phosphorylation and the modulation of the pentose phosphate pathway related to ROS signaling response [31]. CO-induced reinforcement of oxidative metabolism facilitates prostate cancer treatment [32] and neuronal differentiation [33] and also prevents cell death [29] and limits neuroinflammation [34,35]. Revisions on the biological role of CO in homeostasis and cytoprotection are available [36,37].

Therefore, administration of low CO levels holds great therapeutic potential, with considerable ongoing biomedical research and clinical trials [37]. The administration of these low levels of CO in humans is made by inhalation (iCO). However, in preclinical animal studies of many animal models of disease, CO has been administered through prodrugs or CO-loaded materials. The prodrugs are generally named CO-releasing molecules (CORMs), which can be based on metal carbonyl complexes [38], organic molecules [39,40], or materials loaded with CORMs (CORMAs) [41,42]. All these developments around CO delivery methods different from simple inhalation intend to improve the selective targeting of CO to the disease tissues, which is impossible with inhalation whereby CO is transported in the blood stream without any tissue selectivity, either as dissolved gas in the plasma or as COHb. In all cases, in vivo animal studies with CORMs and/or iCO therapy have used measurements of COHb as a control of exposure and toxicity since, in humans, the latter is absent below 15% COHb with very rare, alleged exceptions. In the case of CORMs and CORMAs, efficient targeting is expected to reduce COHb to very low values just like what happens when HO-1 is activated at the site of disease. In this sense, a clear and rapid COHb rise is an unwanted event. Notwithstanding, some experimental evidence suggests that COHb is not toxic and could instead possess protective activities. Thus, the cytoprotection elicited by CO may be due not only to its direct effect on cells but may also imply COHb signaling.

## 3. Hemoglobin in Red Blood Cells

Erythrocytes or red blood cells (RBCs) correspond to about one quarter of human cells. They are generated and develop in bone marrow and have a rapid cellular population turnover with a lifetime of 100 to 120 days in circulation. These cells lack nuclei and most of the cellular organelles in order to maximize O_2_ transport by presenting great amounts of Hb (~98% of total protein). Therefore, cellular function is mainly controlled by protein–protein interactions and allosteric and post-translational modifications [43].

Because of the high O_2_ tension and abundant iron content, RBCs are prone to generate reactive oxygen species (ROS), which trigger oxidative stress and cell damage. In fact, Hb can suffer slow autooxidation, reducing O_2_ to the superoxide anion (O_2_^∙−^) and oxidating Fe^2+^ to Fe^3+^, thus leading to the formation of MetHb, which is unable to bind O_2_ (Figure 1). The auto-oxidation of Hb is a slow process, but can be accelerated under hypoxia [44]. Likewise, dismutation of O_2_^∙−^ produces hydrogen peroxide (H_2_O_2_) which via the Fenton reaction oxidizes Fe^2+^ to Fe^3+^ along with hydroxyl radical (HO^∙^) production, functioning as an amplification loop. In addition, Hb and MetHb can also function as pseudo peroxidases converting H_2_O_2_ into H_2_O under uncontrolled pathological conditions [45]. In particular, Fe^3+^ in MetHb is a dangerous molecule that can react with H_2_O_2_ leading to Fe^4+^ ferrylHb formation, which is a powerful oxidant [46] (Figure 1). Endothelial vessel cells and macrophages produce NO which can cross membranes reaching RBC. Once there, NO can oxidize Hb to MetHb, generating nitrate, as well as react with a superoxide, forming peroxinitrite, which is a highly reactive and tissue damaging molecule [44].

## 4. RBC Anti-Oxidant Machinery

In order to avoid the oxidative processes mentioned above, the RBC environment must be very reducing with strong antioxidant systems able to maintain Hb at the functional reduced Fe^2+^ state. Accordingly, the RBC antioxidant machinery is constituted by enzymes such as catalase, superoxide dismutase, glutathione peroxidase, and peroxiredoxin-2 enzymes [44]. Catalase and glutathione peroxidase are the main antioxidant enzymes involved in the maintenance of a reducing environment in RBCs [47]. Moreover, high levels of intracellular reduced GSH and ascorbic acid are key elements for maintaining iron at its reduced state [48]. In RBCs, ascorbic acid (AA) can reach concentrations as high as 2 mM. In response to an oxidant, AA is oxidized to dehydroascorbic acid (DHA), which must be recycled by reacting with reduced glutathione (GSH). Likewise, AA is also a critical reducing molecule in plasma, where its oxidation product, DHA, is transported into RBCs via GLUT-type D-glucose transporters for reduction into AA by GSH [46]. RBCs do not contain mitochondria, with glycolysis being their main source of ATP. Glycolysis is linked to the pentose phosphate pathway (PPP), which is the key metabolic pathway for the generation of the reducing factor NADPH needed for GSSG reduction into GSH [49] (Figure 2). Alternatively, Ogasawara and colleagues also demonstrated that in response to *tert*-butylhydroperoxide treatment (oxidative stress) the reduction of MetHb to Hb was not supported by increased levels of PPP but by glycolysis and NADH generation. In fact, there is an increase in glucose metabolism via glycolysis, generating more pyruvate and NADH via glyceraldehyde-3-phosphate dehydrogenase (GAPDH) [50]. In conclusion, glucose metabolism can be antioxidant in RBCs through different metabolic pathways (Figure 2).

Finally, the Hb protein itself also presents antioxidant functions. Whenever its β93 cysteine (β93Cys) residue is changed for alanine, there is an increase in ROS generation in response to stressful stimuli such as inflammation in mice [51].

## 5. Cell-Free Hemoglobin

The presence of cell-free Hb in the plasma occurs following hemolysis with the leakage of Hb from RBCs. Pathological hemolysis is a consequence of hemolytic diseases, namely hemorrhage, sickle cell disease, autoimmune induced anemia, infection-induced anemia, thalassemia, massive blood transfusion, among others [52,53]. Under pathological hemolysis, the toxicity of cell free-Hb and heme is not limited to blood vessels but also affects organ parenchyma. In contrast, under physiological conditions, vascular hemolysis also occurs as a consequence of normal RBC turnover. It is well established that the concentration of intracellular Hb (inside erythrocytes) is 10 mM, while the extracellular Hb concentration is around 2 μM in healthy humans [54]. Nevertheless, a study performed in 1959 calculated that about 10% of total RBC breakdown occurred in blood vessels [55]. Despite these controversial data, one can speculate that the systems for scavenging cell-free Hb are quite efficient (please see Section 6).

Inside RBCs, Hb is an α2β2tetramer protein with reduced Fe^2+^ and an efficient cellular machinery for the maintenance of a reducing environment. In contrast, in an extracellular environment, tetramer Hb is in balance with αβdimer Hb, with a predominant dimer state depending on the cell-free Hb concentration [56]. αβdimer Hb is smaller, which facilitates its extravasation into perivascular tissue, increasing oxidative stress in vulnerable organ parenchyma (the kidneys and liver) [56]. In addition, there is NO-induced oxidative stress by oxidizing oxyHb-Fe^2+^ to Hb-Fe^3+^, which easily releases the reactive ferric protoporphyrin-IX group (heme group). In fact, heme is a strong prooxidant and pro-inflammatory molecule, exacerbating tissue damage [56]. Cell-free Hb-Fe^2+^ also scavenges endogenous NO produced by endothelial cells. Thus, NO does not reach smooth muscle cells, and vasodilation is inhibited [57,58]. In fact, some hemolytic diseases are associated with hypertension since the lack of NO promotes vasoconstriction [59] (Figure 3). Interestingly, cell-free methemoglobin can suffer auto-reduction, generating cell-free Hb-Fe^2+^, which in turn scavenges NO and promotes vasoconstriction [60].

In vascular endothelial cells, cell-free Hb induces oxidative stress, upregulation of heme-oxygenase and ferritin, and cell injury, with the effect being much more pronounced whenever MetHb is present [61,62]. Cell-free ferrylHb (Hb-Fe^4+^) triggers inflammation and permeability of vascular endothelial cells, changing actin cytoskeleton organization and increasing the expression of pro-inflammatory genes [63]. Likewise, interaction between cell-free Hb and atheroma lipids promotes an amplification loop of vessel lesions, involving oxidative stress and inflammation. In fact, oxidized LDL (low density lipoprotein) and lipids derived from atherosclerosis patients facilitates oxidation of MetHb to ferrylHb [64]. Likewise, treatment with ascorbic acid can eventually decrease the cell-free-Hb-induced permeability and lesions in the endothelium [65]. Finally, being a pro-inflammatory molecule, ferrylHb also activates macrophages, promoting a proinflammatory environment in tissues [66].

Great amounts of plasma are filtered by the kidneys daily; therefore, cell-free Hb and its metabolites are highly nephrotoxic, inducing oxidative stress, apoptosis, and inflammation [67,68], which can lead to acute and eventually chronic kidney disease [69,70].

Cerebral microvascular endothelial cells along with astrocytes and pericytes form the blood-brain barrier (BBB), which is a key structure with higher selectivity and limited permeability for the maintenance of brain homeostasis as a sanctuary organ [71]. Intracerebral or subarachnoid hemorrhage are the main causes of cell-free Hb affecting the brain, although vascular hemolysis with systemic circulating cell-free Hb also disturbs the BBB and cerebral parenchyma [72]. Systemic exposure to Hb alters the organization of tight junction proteins and promotes lipid peroxidation and iron deposition in the BBB, leading to its permeabilization [73]. Moreover, rat intracerebral injection of Hb solution promotes BBB leakage by inducing endothelial and inducible NO synthase activity and NO production, which in turn alter tight junction proteins’ organization [74]. Similarly, in hemorrhagic stroke models based on cerebral Hb injection, BBB permeability occurs by peroxinitrite formation following Hb’s reaction with NO [75] or by activation of matrix metalloproteinase-9 (MMP-9) inducing apoptosis in the EC of the BBB [76].

Hb dimer and free heme are small molecules that can easily cross endothelial barriers and infiltrate into tissues; thus, their deleterious effect can also reach other compartment than just the blood or endothelium [56,77]. In particular, in the CNS, cell-free Hb is neurotoxic when reaching the brain parenchyma by triggering neuronal cell death [78] and by inducing neuroinflammation with microglial activation and the release of many proinflammatory cytokines [79,80]. Likewise, Hb breakdown products actively participate in brain injury following intracerebral hemorrhage [81]. The presence of cell-free MetHb in cerebrospinal fluid (CSF) following preterm cerebral intraventricular hemorrhage is associated with pro-neuroinflammatory response with higher levels of TNF-α, IL-1β, and TRL-4 [82]. In astrocytic cell culture, MetHb treatment also increases TNF-α expression, while oxyHb does not [82].

In summary, cell-free Hb is a highly reactive molecule, with it becoming a dimer protein with oxidized iron, that in turn is extremely toxic, with the endothelium being the first target.

## 6. Haptoglobin and Hemopexin

During evolution, nature has developed systems to protect organisms from cell-free Hb and free heme: haptoglobin and hemopexin, respectively. Haptoglobin (Hp) is an α2- sialoglycoprotein and is the primary Hb-binding protein in the plasma [83]. Hp binds to dimers of Hb, which is rapidly formed following Hb’s release from RBCs [84]. Under physiological conditions, circulating Hp scavenges free Hb originating from normal RBC turnover, thus inhibiting Hb-mediated toxicity. Then, the complex Hb-Hp interacts with CD163 receptors of monocytes/macrophages for its internalization and further clearance in the lysosomes, with upregulation of heme-oxygenase (HO) for heme degradation [85]. Whenever excessive and pathological hemolysis occurs, Hp is depleted in the plasma, and cell-free Hb leads to oxidative stress and endothelium and tissue lesions [83]. Interestingly, there are many studies targeting the levels and the genotype of circulating Hp as a potential biomarker for disease severity, in particular in hemorrhagic diseases [86,87].

In contrast, free heme is first bound to albumin and then is transferred to hemopexin (Hpx). The complex heme-Hpx is primarily eliminated in hepatocytes after its internalization via LDL-receptor-related protein 1 (LRP1) [83]. In fact, the presence of hemopexin reduces endothelial oxidative stress and brain lesions following atherosclerosis [88] and intracerebral hemorrhage [89]. Interestingly, α1-microglobulin is a heme-binding protein also able to scavenge free heme following hemorrhage in atherosclerosis lesions. In fact, the pro-inflammatory ferrylHb and free heme promote expression of α1-microglobulin [90].

## 7. Is Carboxyhemoglobin (COHb) Toxic?

Since the ancient era, coal fumes are known to be toxic. In the 18th century, the concept of a gas being produced by combustion was established, and a more systematic characterization of gas intoxication in humans was carried out. Still in 1796, this gas was described to have more affinity to “animal fibers” than oxygen, but only in 1800 was CO’s structure found [11]. Claude Bernard discovered that CO displaces oxygen leading to body hypoxia in 1846, and in the 20th century, Haldane and colleagues carried out a quantitative study on CO’s affinity to hemoglobin. For an interesting and complete historical review about CO, please see [11]. With a long role in human history, carbon monoxide (CO) is widely known to the public as a silent killer due to its high affinity to Hb, forming carboxyhemoglobin (COHb). CO presents 240-fold greater affinity for Hb than O_2_, limiting O_2_ delivery into tissues, which can lead to systemic hypoxia [11,91]. Interestingly, CO partially binding to Hb increases the affinity of the remaining O_2_ molecules to Hb, limiting their release into the tissues [92].

With the high affinity of CO to Hb, it is expectable that the storage of endogenously produced CO occurs under the form of COHb. Under physiological conditions, CO is continuously produced by the catabolism of free heme corresponding to more than 90% of endogenous CO production, with the remaining CO coming from lipid and xenobiotic metabolism [93]. Thus, an adult generates 20–50 mL of exhaled CO gas per day under physiological conditions. On the other side, 10–15% of normal erythrocytes’ turnover occurs by vascular hemolysis, leaking hemoglobin (Hb) and free heme into the plasma [55,56]. Thus, endogenous CO is scavenged by Hb, leading to basal COHb production. Indeed, the human basal levels of circulating COHb in a non-smoker subject are 0.1 to 2%, depending on the environmental conditions [11,94]. Moreover, under pathological conditions (in response to inflammation, oxidative stress, hypoxia, etc.), the levels of HO-1 expression increase along with CO generation. Therefore, elevation in COHb circulating levels also occurs.

Since in biology nothing is produced without purpose, it is hypothesized that COHb is not an inert or waste by-product of CO metabolism. Rather, COHb might present an active biological role and thus, COHb could be considered as an integral part of the cytoprotective HO/CO axis previously described. In the next sections, literature data will be presented to support this hypothesis.

## 8. COHb Is Not a Measure of CO Toxicity

In biological systems, CO binds to heme proteins, changing their function. In fact, CO-induced intoxication is usually associated with deficient systemic oxygen delivery as previously mentioned, while cytotoxicity is mostly associated with CO binding to cytochrome c oxidase which in turn inhibits mitochondrial respiration leading to ATP deprivation and oxidative stress [95].

For decades, the levels of circulating COHb have been considered a measure of CO intoxication. Nevertheless, a great variety of COHb concentrations are associated with fatal CO intoxication, ranging from 3% to 98% [96]. It is important to note that with such a wide range of values, its statistical significance is almost meaningless. Additionally, the highest reported value of COHb levels in an individual who survived is 73% [97], reinforcing the point that COHb does not accurately correlate with CO poisoning. Thus, COHb should only be used as a confirming biomarker of recent CO exposure and not as a measure of CO poisoning severity [98].

Moreover, in 1975, an important in vivo study on systemic CO toxicity showed no correlation between COHb levels and CO toxicity [99]. When a group of dogs was exposed to 13% of CO gas by inhalation, COHb levels reached 54% to 90%, and all animals died within 1 h. In the second group, the dogs received a transfusion of CO-loaded blood, leading to a final value of 80% COHb. In this case, all animals survived [99]. These results indicate that COHb levels cannot be taken as a measure or predictor of systemic CO toxicity. If anything, they show that exogenously administered COHb is not toxic and that, in the case of CO inhalation, toxicity is most likely due to free CO. Accordingly, in 2021, Mao and colleagues developed an accurate and sensitive colorimetric assay for CO quantification in tissues and blood following CO inhalation [100]. This assay is based on the synthetic supramolecule hemoCD1, synthesized by Kitagishi and colleagues, composed of an iron(II)porphyrin, meso-tetrakis(4-sulfonatophenyl) porphinatoiron-(II), and a per-*O*-methylated β-cyclodextrin dimer, which binds to CO with an affinity higher than that of Hb [101]. Following CO gas inhalation at 400 ppm, the blood COHb concentration linearly increases over time, while CO levels in different tissues (the liver, lungs, cerebrum, cerebellum, and heart) rapidly reach a plateau below the tissue’s saturation capacity. These data suggest that COHb is formed to prevent toxic accumulation of free CO in tissues, ultimately resulting in protection [100].

In summary, COHb in circulation is not toxic and circulating COHb concentration must not be considered a measure of systemic CO toxicity. Likewise, CO-induced toxicity may be mostly associated with CO gas which binds to cellular heme proteins, in particular cytochrome c oxidase. Nevertheless, one can also envisage a scenario where COHb would deliver CO to other heme proteins depending on its affinity, but this needs further investigation.

## 9. How Can COHb Be (cyto)Protective?

Several hypotheses can be considered for endogenous COHb to have a biological role that can be systemically protective and/or cytoprotective.

### 9.1. COHb Formation Is a Protection Mechanism against Cell-Free Hb Oxidation and Toxicity

Upon hemolysis, cell-free Hb is easily oxidized from Hb-Fe^2+^ into Hb-Fe^3+^ (methemoglobin, MetHb) or Hb-Fe^4+^ (ferrylHb), which in turn promote oxidative stress, lipid peroxidation, inflammation, and BBB disruption [56,72]. Cell-free COHb is more stable and less toxic than cell-free Hb [102]. For example, in vitro studies mimicking plasma conditions demonstrated that CO reduces Hb-Fe^3+^ into carboxyHb-Fe^2+^ in the presence of peroxide as an electron donor [102]. Likewise, in a model of hemolytic malaria (*P. berghei*-infected mice), CO gas exposure decreased the levels of circulating MetHb and free heme, which limits oxidative stress [103]. Furthermore, the Kirklareli mutation (H58L) in human Hb promotes mild anemia because the mutated Hb is susceptible to high auto-oxidation [104]. Interestingly, smokers carrying the mutation are protected against anemia when compared to non-smokers. This is due to the fact that the CO derived from cigarette smoke tightly binds to the mutated Hb and prevents its oxidation into MetHb [104].

Administered as a mouse intraperitoneal injection, hemoCD binds first to free endogenously produced CO and then captures CO from circulating COHb. Thus, hemoCD generates a depletion of endogenous CO, which in turn upregulates liver HO-1 expression to compensate for the lack of CO. In fact, after losing CO, plasma oxyHb is rapidly oxidized into MetHb, releasing free heme which triggers HO-1 expression [101].

Cell-free or RBC COHb levels may also represent novel potential biomarkers of disease. In fact, high levels of cell-free Hb in human plasma have been found to be associated with worse prognosis in sepsis [105]. Likewise, in intensive care units, COHb is a biomarker for hemolysis. In fact, hemolysis releases cell-free Hb, which is eliminated by macrophagic phagocytosis, increasing the expression of HO-1 for heme degradation which increases CO, thus forming COHb [106]. Thus, COHb holds potential as a diagnostic or prognostic circulating biomarker.

Altogether, these data point to another CO mechanism of action: the generation of COHb protecting against cell-free Hb’s deleterious effects (Figure 3).

### 9.2. COHb Is an Antioxidant

In vitro, the exposure of red blood cells (RBCs) to CO promotes a great increase in the levels of intracellular reduced GSH, from about 2 mM to 3 mM [107]. CO does not promote glycolysis but partially increases the pentose phosphate pathway. Nevertheless, the main origin of this great increase in GSH levels is Hb de-glutathionylation, meaning the release of glutathione from cysteine residues, in particular Cys93 and Cys112 [107]. COHb formation alters Hb conformation and releases GSH, which in turn plays a key antioxidant role. Therefore, COHb formation in RBCs due to the presence of CO reinforces their antioxidant defenses (Figure 4).

Using an in vivo rat model of hemorrhagic shock, transfusion of RBCs or vesicles containing Hb exposed to CO gas, that is containing COHb, showed better systemic parameters (arterial blood pressure, lactate, or PO_2_ and PCO_2_) than whenever a transfusion was performed with RBCs or vesicles containing Hb [108]. In addition, COHb vesicles or CO-RBCs decreased oxidative damage in the lungs and liver when compared to Hb vesicles or RBC transfusion. Moreover, the circulating COHb levels decreased from 26–39% immediately after transfusion to 3% after 6 h [108]. RBC transfusion is the gold standard treatment for massive hemorrhage. Despite this, this therapy is associated with great hepatic oxidative stress and a decrease in cytochrome P450 activity. Whenever RBCs exposed to CO (CO-RBCs) were used for transfusion, there was a reversion of the deleterious effects of RBCs, with a decrease in nitric oxide and free heme levels, an increase in cytochrome P450 activity, and lower oxidative stress biomarkers [109,110]. Likewise, at a late stage after resuscitation, CO-RBCa also limited the increased levels of the plasma pro-inflammatory markers IL-6 and TNF-α [111]. Finally, in a rat model of acute kidney injury triggered by traumatic rhabdomyolysis and hemorrhage, transfusion of CO-RBCs was revealed to be renoprotective, avoiding the oxidative stress induced by free heme [112]. In summary, transfusion of CO-RBCs confers systemic protection against oxidative stress in the context of hemorrhagic disorders. It should also be tested in other models, including organ transplant or blood transfusion after surgery.

### 9.3. COHb May Present Other Protective Functions

The most frequently used concentration of CO gas inhalation in animal models is 250 ppm, which promotes protection in several different tissues and pathologies [93]. Depending on the exposure time and the animal model, 250 to 500 ppm leads to circulating COHb levels between 5 and 30% [113]. Thus, the protective effect may be due to free CO reaching tissues or due to circulating COHb or even both.

In addition, ALF186 is a CO-releasing molecule (CORM) that readily delivers its CO load to RBC, reproducibly raising COHb within minutes after mouse injection, with most of the released CO binding to COHb [114]. Interestingly, in vivo i.v. administration of ALF186 protects retinal ganglion cells against ischemia and reperfusion [115]. Because practically all CO molecules released from ALF186 bind to Hb, one can speculate on a direct effect of COHb in tissues, in this case a protective effect in the retina [115]. Nevertheless, this experiment should be repeated with COHb administration instead of CO to confirm the role of COHb.

Finally, another clue indicating that COHb might have a protective role is disclosed by smokers or the phenomenon called the smoking paradox. In fact, smokers present on average higher circulating COHb levels than non-smokers, going from 2% to 10% [116,117]. Despite being extremely injurious and harmful, epidemiological data show that smokers present lower levels of incidence of Parkinson’s disease [118], pre-eclampsia [119], and some skin cancers [120], among other diseases [11]. These protective effects have been attributed to free CO; however, it can be speculated that the higher levels of COHb may also play a role. Of course, molecules other than CO may also be involved in the smoking paradox, since tobacco smoke has a complex composition.

## 10. Impact and Future Perspectives

The research about COHb as a protective molecule opens a new paradigm in the CO biology field. For 20 years, scientists have searched for the underlying CO mechanisms of cytoprotection, without paying attention to COHb. Moreover, the development of CORMs has always been guided by the idea of avoiding extensive COHb formation to keep the CORM structure intact until it reaches the tissues in need to deliver CO locally. It is urgent now to expand scientific knowledge on the physiological role of COHb, which may also open new avenues for therapy development.

In fact, COHb presents advantages as a therapeutic agent since it is an endogenous molecule that exists permanently in organisms, meaning that our body is fully adapted to it. The advantages of using COHb instead of CO gas or any type of CORMs are obvious. In the former case, the opposition to the use of CO gas due to the widespread, deeply rooted perception of its potential toxicity would be removed. In the latter case, beyond CO, the added risks posed by xenobiotic molecules from scaffolds, carriers, or metabolites would be absent. Many blood substitute products based on Hb have been developed to replace blood and deliver oxygen into tissue. Following this paradigm, PEGylated COHb emerges as a potential therapeutic agent against hemorrhagic and ischemic stroke since it delivers CO and O_2_ into tissues [121].

## Figures and Tables

**Figure 1 antioxidants-12-01198-f001:**
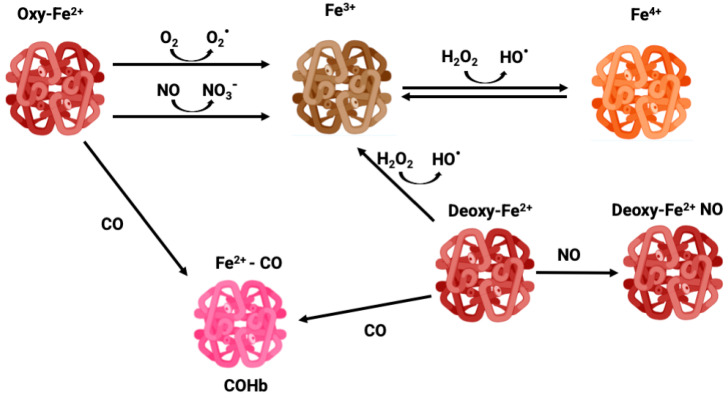
Different oxidative states of iron—oxyhemoglobin, methemoglobin, carboxyhemoglobin, and ferryl-hemoglobin.

**Figure 2 antioxidants-12-01198-f002:**
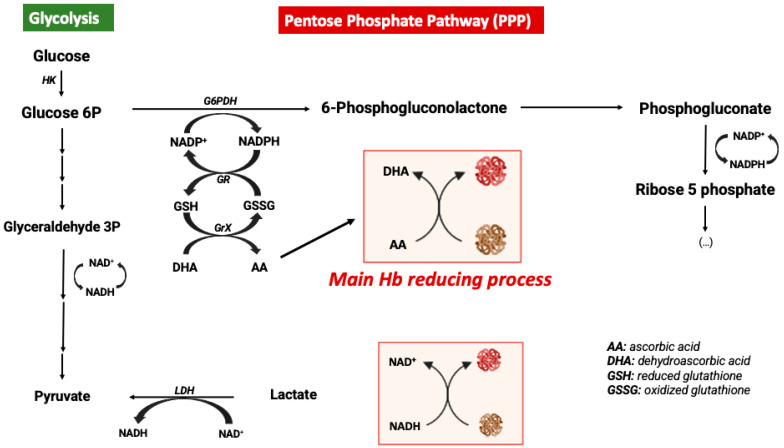
Metabolic modulation of hemoglobin redox state.

**Figure 3 antioxidants-12-01198-f003:**
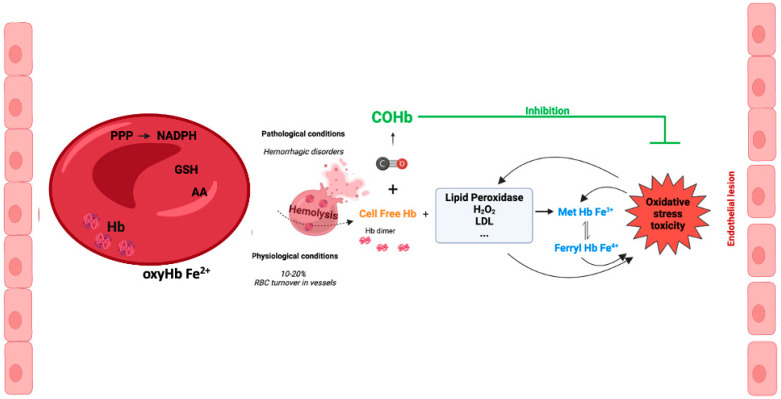
Cell-free hemoglobin and cell-free carboxyhemoglobin—metabolism and reactions.

**Figure 4 antioxidants-12-01198-f004:**
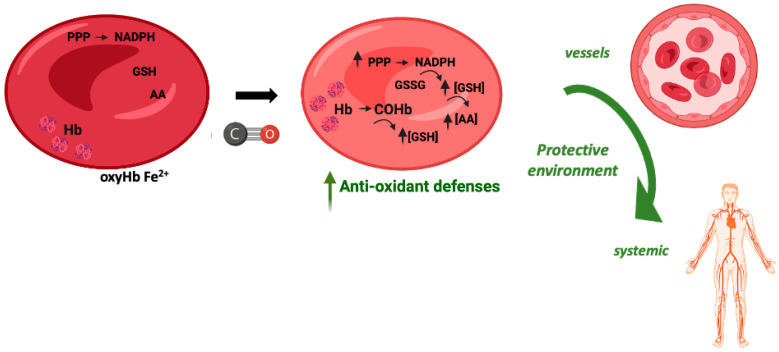
Potential cytoprotective and antioxidant effects of carboxyhemoglobin.

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
