# Peer review of "Carboxyhemoglobin (COHb): Unavoidable Bystander or Protective Player?"

_antioxidants, 2023, doi:10.3390/antiox12061198_

Round 1
Reviewer 1 Report
The paper describes two main premises: (i) there is no direct correlation between COHb levels and CO toxicity; and (ii) COHb seems to have a direct cytoprotective and antioxidant role in erythrocytes and in hemorrhagic models in vivo. In my opinion it was well done. I have only marginal remarks: All abbreviations are explained. Please, also for TCA.
Toxicity of free Hb should also included nephrotoxicity as one of the main aspects.
It would be nice to read also something about hemoglobin based oxygen carriers (HBOC), but mybe it is better to describe it in a separate review.
Author Response
The paper describes two main premises: (i) there is no direct correlation between COHb levels and CO toxicity; and (ii) COHb seems to have a direct cytoprotective and antioxidant role in erythrocytes and in hemorrhagic models in vivo. In my opinion it was well done. I have only marginal remarks: All abbreviations are explained. Please, also for TCA.
It is now included in the manuscript.
Toxicity of free Hb should also be included nephrotoxicity as one of the main aspects.
This is a very important suggestion since kidneys have a great interaction with plasma containing cell free Hb and its metabolites. Thank you. It is now included in the new version of the manuscript.
It would be nice to read also something about hemoglobin based oxygen carriers (HBOC), but mybe it is better to describe it in a separate review.
Hemoglobin based oxygen carriers (HBOC) are a very interesting topic but indeed it is not in the scope of the present review.

Reviewer 2 Report
Carbon monoxide has been the crossfire of two views in biology, especially in human pathophysiology. The first view represents the toxic aspects of CO, the second one accumulates data to prove, that this unique gas can have beneficial effects as well. This review helps us to solve this contradiction. The center of their focus is the administration of CO to the animals, in certain cases to cells, and in the future for humans in strict conditions. The administration routes are the inhalation of CO, or CO releasing agents, and the second is the CO-Hb (CO-red blood cell Hb) transfusion. This is greatest value of this work.
I would suggest a phenomenon for the authors, which should be discussed in details. Carbon monoxide has a special and very significant effect on free heme release from hemoglobin. It is the same as cyanide. If metHb is treated with CO, heme is locked to the globin chains, heme is not released from the protein. It can be checked by heme oxygenase 1 induction in cell culture. We know that free heme is extremely toxic acutely, so this locking effect has importance. This is the significance of CO-red blood cell transfusion. Inhalation may have similar effects, but myoglobin-, cytochromes-, mitochondrial toxicity covers its protective nature.
Minor suggestions are the followings.
CO-Hb should have bright red color, in clinics the color of the patients is bright red.
Neuro-inflammation should be combined in one part, if it is possible.
MetHb has a peroxidase activity. I think it is more a pseudo-peroxidase activity.
Hemolysis can occur in the extravascular space, for example, in hemorrhages, in brain, in tissues, inside of an atherosclerotic vessel wall. We can’t neglect this type of red blood cell lysis. It should be discussed.
FerrylHb is an extremely important pro-inflammatory Hb derivative, because it polarizes macrophages for a pro-inflammatory cell type, and also it is the origin of many pro-inflammatory peptides. It would be nice to see these facts in this review.
Although hemopexin is extremely important plasma protein, but alpha-1-microglobulin is also an effective extracellular heme binder, which is induced by FerrylHb in many resident cells of vascular walls.
This is nice review, some newer references would increase it value. I hope, the authors will accept my suggestions.
Minor editing of English language required.
Author Response
Carbon monoxide has been the crossfire of two views in biology, especially in human pathophysiology. The first view represents the toxic aspects of CO, the second one accumulates data to prove, that this unique gas can have beneficial effects as well. This review helps us to solve this contradiction. The center of their focus is the administration of CO to the animals, in certain cases to cells, and in the future for humans in strict conditions. The administration routes are the inhalation of CO, or CO releasing agents, and the second is the CO-Hb (CO-red blood cell Hb) transfusion. This is greatest value of this work.
I would suggest a phenomenon for the authors, which should be discussed in details. Carbon monoxide has a special and very significant effect on free heme release from hemoglobin. It is the same as cyanide. If metHb is treated with CO, heme is locked to the globin chains, heme is not released from the protein. It can be checked by heme oxygenase 1 induction in cell culture. We know that free heme is extremely toxic acutely, so this locking effect has importance. This is the significance of CO-red blood cell transfusion. Inhalation may have similar effects, but myoglobin-, cytochromes-, mitochondrial toxicity covers its protective nature.
Our team has been working for years in the CO Biology field. COHb is a new subject, which we are actually very interested in working with and developing new research lines. We are really grateful for your comments and suggestions that have opened new hypothesis of how CO could limit toxicity of free heme. The fact that CO locks MetHb and avoids heme release is very interesting fact (and new for us). Unfortunately we are not able to find references with this information. Could you please send us those references. It would be of great importance to mention this fact in the Review.
Minor suggestions are the followings.
CO-Hb should have bright red color, in clinics the color of the patients is bright red.
The COHb color in the figures have been altered
Neuro-inflammation should be combined in one part, if it is possible.
Thank you for this suggestion. In the new version of the manuscript neuroinflammation is now added and discussed (highlighted in yellow).
MetHb has a peroxidase activity. I think it is more a pseudo-peroxidase activity.
It has been changed in the new version of the manuscript – highlighted in yellow
Hemolysis can occur in the extravascular space, for example, in hemorrhages, in brain, in tissues, inside of an atherosclerotic vessel wall. We can’t neglect this type of red blood cell lysis. It should be discussed.
It is now better presented and discussed hemorrhage and the fact that cell free Hb also damage organ parenchyma.
FerrylHb is an extremely important pro-inflammatory Hb derivative, because it polarizes macrophages for a pro-inflammatory cell type, and also it is the origin of many pro-inflammatory peptides. It would be nice to see these facts in this review.
It is now mentioned in the new version of the manuscript.
Although hemopexin is extremely important plasma protein, but alpha-1-microglobulin is also an effective extracellular heme binder, which is induced by FerrylHb in many resident cells of vascular walls.
Thank you for this interesting information. It is now added in the manuscript.
This is nice review, some newer references would increase it value. I hope, the authors will accept my suggestions.
Because it is a novel manner to approach COHb biology, we have been deeply searching data about COHb. Data that are not the “normal” toxic information about COHb and CO, nor the classical protective role of endogenous CO. Thus, some old and very interesting references were found.
Indeed, your comments have enriched the Review. Thank you again!

Reviewer 3 Report
The paper entitled “Carboxyhemoglobin (COHb): unavoidable bystander or protective player?” presents an interesting perspective that caboxyhemoglbin (COHb) plays a beneficial role in normal health and therapeutics. The authors provide a good background on CO and hemoglobin and cite several interesting examples supporting their view. There are a few places where some information is not correct and, in general, a more practical, balanced discussion would be helpful.
1. The notion that COHb protects against oxidative stress by Cell-free Hb could be misleading and needs more thought. It is known that NO reacts preferentially with cell-free Hb compared to erythrocytic Hb. Thus, just a few micromolar cell-free Hb can significantly scavenge NO even in the background of 10 millimolar erythrocytic Hb. This is not the case for CO. There is no preferential reactivity of CO with cell-free Hb compared to that in the red cell. This is due to the substantially slower rate of reaction of CO with Hb compared to NO, so that diffusion is not a factor. Thus, one would need to have high overall percentage of COHb to make a difference in the percentage of cell-free Hb that is CO bound. To accomplish this, one would have to administer a lot of CO.
2. The statements on NO reactions with Hb on lines 55-57 are incorrect. OxyHb reacts with NO to form MetHb and nitrate (not nitrite). NO binds to deoxyHb forming iron nitrosyl Hb. This reaction does not lead to methemoglobin formation. NO, upon oxidation to NO+, can binds to cysteine on the BETA chains (not alpha). SNO-Hb as it is referred to is due to the mdofication ate the beta93 cysteine. Corrections to Figure one are also needed. In that figure, NO2 is presumably supposed to be nitrite. However, there is no minus sign shown so it is actually shown as nitrogen dioxide. Nitrogen dioxide (radical) is quite different from nitrite.
3. The statement that 10-20% of red cells lyse in vessels cannot be right. Importantly, cell-free Hb in healthy individuals is less than 2 micromolar while that is the RBC is about 10 mM – a factor of 5000.
4. In the paragraph starting with line 171, it is stated that most of cell-free Hb is in the dimer form. This would depend on how much cell-free Hb there is. The dissociation constant in the tetramer-dimer equilibrium is about 1 micromolar. In addition, cell-free Hb is not predominantly metHb – see for example Reiter et al Nat Med 2002. If the cell-free Hb were not ferrous, it could not scavenge NO. A paper in Transfusion (TRANSFUSION 2013;53:3149-3163) showed that methemoglobin infused into dogs is reduced to oxyhemoglobin.
5. The authors state a premise that there is no correlation between COHb levels and CO toxicity. They raise some good points supporting this premise where, for example, high levels of COHb are measured with low toxicity. However, this discussion may benefit from a more thorough elucidation of what causes pathology due to CO poisoning. This reviewer understands that interference with mitochondrial respiration due to CO binding to cyt c oxidase is a major contributor. COHb could deliver CO to mitochondria. Thus, an acute measure of COHb may not correlate with pathology, but enough COHb for long enough a time would likely cause pathology. That said, the study quoted where there was 80% COHb in dogs with no mortality is intriguing.
NA
Author Response
The paper entitled “Carboxyhemoglobin (COHb): unavoidable bystander or protective player?” presents an interesting perspective that caboxyhemoglbin (COHb) plays a beneficial role in normal health and therapeutics. The authors provide a good background on CO and hemoglobin and cite several interesting examples supporting their view. There are a few places where some information is not correct and, in general, a more practical, balanced discussion would be helpful.
We are really grateful for your review and comments. Actually, we have been working for years in the CO Biology field, while COHb is a new field of research for us, an interesting hypothesis that we want to explore. Your comments are so valuable and crucial that we would like to collaborate with you and go deeper and further in the research of a biological role of COHb. Please find our answers and comments bellow:
- The notion that COHb protects against oxidative stress by Cell-free Hb could be misleading and needs more thought. It is known that NO reacts preferentially with cell-free Hb compared to erythrocytic Hb. Thus, just a few micromolar cell-free Hb can significantly scavenge NO even in the background of 10 millimolar erythrocytic Hb. This is not the case for CO. There is no preferential reactivity of CO with cell-free Hb compared to that in the red cell. This is due to the substantially slower rate of reaction of CO with Hb compared to NO, so that diffusion is not a factor. Thus, one would need to have high overall percentage of COHb to make a difference in the percentage of cell-free Hb that is CO bound. To accomplish this, one would have to administer a lot of CO.
You are absolutely right: NO is much more reactive and less chemically stable than CO, reacting with Hb in a much faster manner. Still, CO has the limitation that can only bind to Hb when iron is reduced. Nevertheless, CO endogenous production increases in response to stress being COHb also formed intracellularly or extracellularly in the case of hemolysis. Our aim is to understand/hypothesize whether COHb has a biological role, which is it and how. The purpose of the review is not to claim that exogenous/pharmacological administration of CO will be protective via COHb formation, the aim is to disclose/discuss any potential biological role. In order to avoid this misunderstanding it is now clearly stated at section #9 and in lines 55 and 56 of the new version of the manuscript.
- The statements on NO reactions with Hb on lines 55-57 are incorrect. OxyHb reacts with NO to form MetHb and nitrate (not nitrite). NO binds to deoxyHb forming iron nitrosyl Hb. This reaction does not lead to methemoglobin formation. NO, upon oxidation to NO+, can binds to cysteine on the BETAchains (not alpha). SNO-Hb as it is referred to is due to the mdofication ate the beta93 cysteine. Corrections to Figure one are also needed. In that figure, NO2 is presumably supposed to be nitrite. However, there is no minus sign shown so it is actually shown as nitrogen dioxide. Nitrogen dioxide (radical) is quite different from nitrite.
Thank you very much for these corrections and remarks. Actually, this is not our area of expertise and it was written based on the studied literature, and indeed there were some mistakes. Because NO is not the purpose of the manuscript, we have eliminated some detailed description about NO interaction and reactions with Hb, such as NO can also be sequestrated by the cysteine residues on the beta-chain of Hb. Text is now correct, and Figure 1 is correct and more complete. Again thank you very much for your crucial corrections.
- The statement that 10-20% of red cells lyse in vessels cannot be right. Importantly, cell-free Hb in healthy individuals is less than 2 micromolar while that is the RBC is about 10 mM – a factor of 5000.
In a study done by Garby et al in 1959, by injection radiolabeled hemoglobin and measuring over time venous blood, it was possible to calculate Hb outfow from plasma and Hb concentration in plasma. Based on this experimental data, authors were able to validate their predictive values based on theory (calculations). They concluded that a small part (slightly over 10%) of hemolysis occurs intravascularly. In any case, we have added into the review manuscript a further discussion about those numbers and the actually known described intra and extracellular concentrations of Hb under physiological conditions.
- In the paragraph starting with line 171, it is stated that most of cell-free Hb is in the dimer form. This would depend on how much cell-free Hb there is. The dissociation constant in the tetramer-dimer equilibrium is about 1 micromolar. In addition, cell-free Hb is not predominantly metHb – see for example Reiter et al Nat Med 2002. If the cell-free Hb were not ferrous, it could not scavenge NO. A paper in Transfusion (TRANSFUSION 2013;53:3149-3163) showed that methemoglobin infused into dogs is reduced to oxyhemoglobin.
It is now clearly stated that tetramer Hb and dimer Hb are in balance and depends on cell free Hb concentration. The description of iron oxidative state in cell free Hb is now correct, as well as the NO ability to bind to Hb. The Transfusion paper is very interesting and was introduced in the manuscript. Again thank you for your comments and enriching the manuscript content and discussion.
- The authors state a premise that there is no correlation between COHb levels and CO toxicity. They raise some good points supporting this premise where, for example, high levels of COHb are measured with low toxicity. However, this discussion may benefit from a more thorough elucidation of what causes pathology due to CO poisoning. This reviewer understands that interference with mitochondrial respiration due to CO binding to cyt c oxidase is a major contributor. COHb could deliver CO to mitochondria. Thus, an acute measure of COHb may not correlate with pathology, but enough COHb for long enough a time would likely cause pathology. That said, the study quoted where there was 80% COHb in dogs with no mortality is intriguing.
The CO’s cytotoxicity is now mentioned in the manuscript, in particular to its inhibition of mitochondrial respiration and cytochrome c oxidase binding effect. Several papers and studies were quoted to try to point to a new idea of COHb role. One of them is the study with COHb transfusion in dogs, which is indeed surprising but very interesting. If there is interest in the scientific community in deeply studying physiological and pathological role of COHb, this experiment should be repeated, eventually in other models.
